# Machine-Learning-Based Fine Tuning of Input Signals for Mechano-Tactile Display

**DOI:** 10.3390/s22145299

**Published:** 2022-07-15

**Authors:** Shuto Yamanaka, Tatsuho Nagatomo, Takefumi Hiraki, Hiroki Ishizuka, Norihisa Miki

**Affiliations:** 1Department of Mechanical Engineering, Keio University, Yokohama 223-8522, Kanagawa, Japan; shuto.aquarius1997j29@gmail.com (S.Y.); tatsuho19950307@keio.jp (T.N.); 2Faculty of Library, Information and Media Science, University of Tsukuba, Tsukuba 305-8550, Ibaragi, Japan; hiraki@slis.tsukuba.ac.jp; 3Graduate School of Engineering Science, Osaka University, Toyonaka 560-8531, Osaka, Japan; ishizuka@bpe.es.osaka-u.ac.jp

**Keywords:** tactile display, machine learning, inverse problem, tactile perception, mechano-tactile display, piezoelectric, encoding and presentation

## Abstract

Deducing the input signal for a tactile display to present the target surface (i.e., solving the inverse problem for tactile displays) is challenging. We proposed the encoding and presentation (EP) method in our prior work, where we encoded the target surface by scanning it using an array of piezoelectric devices (encoding) and then drove the piezoelectric devices using the obtained signals to display the surface (presentation). The EP method reproduced the target texture with an accuracy of over 80% for the five samples tested, which we refer to as replicability. Machine learning is a promising method for solving inverse problems. In this study, we designed a neural network to connect the subjective evaluation of tactile sensation and the input signals to a display; these signals are described as time-domain waveforms. First, participants were asked to touch the surface presented by the mechano-tactile display based on the encoded data from the EP method. Then, the participants recorded the similarity of the surface compared to five material samples, which were used as the input. The encoded data for the material samples were used as the output to create a dataset of 500 vectors. By training a multilayer perceptron with the dataset, we deduced new inputs for the display. The results indicate that using machine learning for fine tuning leads to significantly better accuracy in deducing the input compared to that achieved using the EP method alone. The proposed method is therefore considered a good solution for the inverse problem for tactile displays.

## 1. Introduction

Communication using tactile information is expected to be an essential technology for next-generation communication systems. Displays that present tactile information have been extensively studied [1,2,3,4]. Mechano-tactile displays are composed of an array of actuators that physically deform the skin, stimulating tactile receptors to create tactile perception [5,6,7]. To stimulate slow adaptive tactile receptors, the actuators need to generate a displacement larger than 100 μm at frequencies below a few tens of hertz, whereas to stimulate fast adaptive receptors, they need to generate a displacement on the order of a micrometer at frequencies of several tens to hundreds of hertz [8]. Our group has proposed and demonstrated a mechano-tactile display that consists of an array of piezoelectric actuators and a hydraulic displacement amplification mechanism (HDAM) [9]. Piezoelectric actuators have a fast response and a large force but a small displacement. An HDAM amplifies the displacement of piezoelectric actuators by a factor of 10, eliminating their weakness. It allows the proposed mechano-tactile display to stimulate both slow and fast adaptive tactile receptors.

Previous work has focused on the development of tactile displays and the characterization of the presented tactile sensations under given conditions (i.e., the forward problem is mainly discussed). However, in practical applications of tactile displays, the tactile sensations to be presented to users are first decided. Then, the inputs to control the displays to produce the sensations must be deduced (i.e., the inverse problem needs to be solved). In our prior work, we proposed an encoding and presentation (EP) method, which exploits the fact that piezoelectric actuators can also act as piezoelectric sensors [10,11]. First, we prepared samples whose surface textures were to be replicated by the tactile display. Then, the samples were slid over the tactile sensor and an HDAM, and then the piezoelectric devices were deformed and generated electric signals according to the surface textures (encoding). The obtained signals, which were on the order of several tens of millivolts, were amplified to drive the actuators of the tactile display (presentation).

Machine learning is a promising tool for solving inverse problems. If a sufficiently large dataset that associates the control parameters of a display and the resulting tactile perception (i.e., solutions to the forward problem) is available, machine learning can deduce the input or the control parameters for the display that are required to produce the target surface. Osgouei et al. applied machine learning to construct an inverse dynamics model for determining the actuation signals for an electrovibration display from the obtained friction force [12]. Sensory experiments verified the effectiveness of this approach, which had a higher accuracy than that of the record-and-playback method, the concept of which is similar to that of the EP method. Lin et al. used the velocity and force in the x- and *y*-axis (i.e., horizontal) directions while the sample surface was scanned with a finger as the input and the acceleration along the *z*-axis was used as the output for machine learning [13]. The acceleration was converted to an input voltage and used to present the target surface texture. Cai et al. used RGB images of textile surfaces as the input and two-dimensional images derived from the friction coefficient as the output in machine learning based on a generative adversarial network [14]. The network deduced the output signals (voltages) that included the characteristic friction coefficients, which were used to replicate the friction coefficient of the target textile using an electrostatic tactile display.

In the present work, we apply machine learning to deduce the more appropriate input signal for actuators than that which is deduced by the EP approach, i.e., to fine-tune the input signal. The dataset used in machine learning consists of encoded surfaces and subjective perception results obtained from participants. The encoded data are in the form of time-domain waveforms. We design a neural network for learning the encoded data (*V* (*t*)) and the subjective evaluation. Note that we do not use the material properties of the samples for learning. We selected five samples, namely wood, a lint-free wipe (Techno Wipe, Nippon Paper Crecia Co., Ltd., Tokyo Japan), polystyrene (PS) foam, a micropatterned polymer (SU8, Kayaku Advanced Materials, Inc., Westborough, MA, USA), and a paper towel (Kim Towel, Nippon Paper Crecia Co., Ltd., Tokyo, Japan). In the user study, participants recorded whether the presented surfaces were similar to any of the samples. Their answers were five-dimensional vectors ([wood, lint-free wipe, PS form, micropatterned polymer, paper towel]), which we refer to as evaluation vectors. Note that the participants did not have to select only one out of the five samples; they could choose multiple samples (e.g., [1, 0, 0, 0, 1] when they considered the presented surface to be similar to the surfaces of wood and a paper towel) or even none of them ([0, 0, 0, 0, 0]). The input signals deduced by machine learning are used to present the surface textures. We investigate the correlation between a presented surface texture and that of the original sample surface to verify the efficacy of the proposed machine learning approach to deduce the input signal and solve the inverse problem for mechano-tactile displays.

## 2. Materials and Methods

### 2.1. Mechano-Tactile Display for Presentation

In this study, the tactile sensations produced by the target material sample were reproduced by independently driving multiple actuators to stimulate a fingertip. Figure 1 shows the mechano-tactile display developed in our prior work [10,11] and also used in this study. The display consists of a 3 × 3 array of large-displacement actuators composed of an HDAM and a piezoelectric actuator (TAK120320, MKT TAISEI Co., Ltd., Saitama, Japan). The HDAM consists of a chamber that encapsulates an incompressible liquid within highly deformable thin membranes made of latex rubber. The membranes are bonded to the top and bottom of the chamber to seal it. The HDAM has a contact part where a finger makes contact and a drive part where a piezoelectric actuator makes contact. When the piezoelectric actuator is driven, a high-frequency small-displacement vibration of the actuator is input to the drive part via a piston made of poly(methyl methacrylate), the displacement of the drive part is amplified based on the ratio of the cross-sectional area of the HDAM, and a high-frequency vibration with a large amplitude is generated at the contact part. The pitch of the array is 4 mm and the diameter of the protruding parts is 0.44 mm when they are not actuated and becomes larger when actuated. The displacement or the height of the actuators reach 1.0 mm at maximum, which can vary when they are in contact with the fingertip. The fabrication process is described in detail in our prior work [10,11].

When we scan surfaces with a fingertip, we tend to move the fingertip in one dimension. Therefore, in this study, we decided that one row that consists of three actuators was controlled as a unit when the display presented the surface. The three-channel digital signals generated by a personal computer (PC) (Precision M4600 Mobile Workstation, DELL, Inc., Round Rock, TX, USA) were converted to analog signals using a digital-to-analog converter interface (USB-3101FS, Measurement Computing Corporation, Norton, MA, USA). The voltage was amplified by three booster modules (As-904, NF Corp., Kanagawa, Japan). The large-displacement actuators were controlled by LabVIEW (National Instruments Corp., Austin, TX, USA). The actuation time, frequency, vibration amplitude, and vibration time difference between the rows were the control parameters.

### 2.2. Tactile Sensor for Encoding

The large-displacement piezoelectric actuator can be used as both an actuator and a sensor. In the EP method, we exploited this property to encode the object surface (i.e., convert the surface shape into a voltage) [11]. The encoded signal was used to drive the actuators of the mechano-tactile display. The structure of the encoder was the same as that of the tactile display but without the top titanium cover, as shown in Figure 2. This structure is mechanically more reliable than that with a cover, which is particularly crucial when a rough or sharp surface is encoded. The material sample was scanned with the encoder, which deformed the top membrane. The piezoelectric sensors output voltage signals. Note that these signals were amplified when used to drive the actuators to present the encoded surface.

### 2.3. Machine Learning Principle

Keras was used as the deep learning framework.

#### 2.3.1. Multilayer Perceptron

Multilayer perceptron (MLP) is a typical feedforward neural network for approximation problems. It has at least one hidden layer with multiple connected neurons and a set of activation functions and weights. An MLP is a supervised learning algorithm that performs nonlinear mapping of the input and output data. It is a model that can be adapted to nonlinear relationships by applying a nonlinear activation function *h*.
(1)$$y=W_{2}h\left(W_{1}x+b_{1}\right)+b_{2},,$$
where the output y is a function of the input x. $W_{1}$ is the input weight, $W_{2}\;$ is the hidden layer weight, and $b_{1}$ and $b_{2}$ are biases.

#### 2.3.2. Activation Function

An activation function converts the sum of the input signals into an output signal at a neuron of the neural network. In this study, the parametric rectified linear unit (PReLU) function is used, which outputs the input value when the input is greater than 0 and outputs the input value multiplied by α when the input value is less than 0, where α is a parameter that is dynamically determined by learning.
(2)$$h\left(\alpha ,\;x\right)=\{\begin{array}{c} x\;\left(x>0\right)\\ \alpha \;x\;\;\left(x\;\leq 0\right) \end{array}$$

#### 2.3.3. Loss Function

A loss function is a measure of the neural network performance (lower value indicates better performance). During the training of a neural network, the weights and biases are updated to minimize the loss function. In this study, the mean squared error was used as the loss function:(3)$$Loss=\frac{1}{n}\;\sum_{i=0}^{n-1}{\left(y_{i}-t_{i}\right)}^{2},$$
where $y_{i}\;$ is the output of the neural network, $t_{i}$ is the training data, and n is the dimension of the dataset.

#### 2.3.4. Optimizer

The goal of learning is to optimize the neural network by finding the parameters that minimize the loss function. In this study, the Adam algorithm, which requires little memory and is computationally efficient, was used as the optimizer. Adam requires 3 hyperparameters to be set. In accordance with previous work, the values were set to $lr=0.001,\;\beta_{1}=0.9,\;\mathrm{and}\;\beta_{2}=0.999$ [15,16].

### 2.4. Data Collection

This section describes the preparation of the dataset. All experiments were approved by the Research Ethics Committee of the Faculty of Science and Technology, Keio University (2020–2032).

In this work, we used five material samples, namely wood, a lint-free wipe, PS foam, a micropatterned polymer, and a paper towel (see Figure 3). The wood, the lint-free wipe, the PS foam, and the paper towel samples are commercially available and were used as the tactile samples in our prior work [11]. The micropatterned polymer was proposed as the microfabricated tactile sample, which can vary the surface geometry, surface chemistry, and the material properties (stiffness, thermal conductivity, etc.) using microfabrication technologies [11]. First, these five samples were encoded and presented to participants using the mechano-tactile display.

First, five material samples were encoded and presented to participants using the mechano-tactile display. The micropatterned polymer was formed using photolithography to have a striped pattern with a width and a spacing of 1000 μm. The participants were asked whether the presented surface was similar to the surface of the corresponding sample. The encoded signals and the answers from the participants were used as the dataset.

To validate that these five material samples can be sufficiently identified, the following preliminary experiment was conducted. Five participants (three males and two females, aged 20 to 22 years) were asked to trace five material samples with their fingertips while their view of the samples was blocked. The order of the samples was randomized and each sample was traced three times. The participants were then asked which of the five materials they had traced. The percentages of correct answers were 100%, 100%, 100%, 100%, and 96.0% for wood, lint-free wipe, PS foam, micropatterned sample, and paper towel, respectively. These results verified that the samples could be identified through tactile perception.

### 2.5. Encoding of Material Surfaces

The material samples were moved by a linear actuator (SGSP26-100, Sigma Koki Co., Ltd., Saitama, Japan) mounted on a laboratory scissor jack (LJ150, Worthef). The speed of the linear actuator was controlled by a PC. A piezoelectric actuator/sensor was connected to the PC via an analog-to-digital converter interface (NI9125, National Instruments Corp., Austin, TX, USA). The voltage output from the piezoelectric actuator/sensor was converted to digital data and stored on the PC. The encoding process was as follows. The contact part of the HDAM was inflated by 1 mm (the initial state). The material samples to be encoded were mounted on the drive unit of the linear actuator with double-sided tape and then scanned over the HDAM at a speed of 4 cm/s for 0.5 s. In case of the micropatterned sample, the scanning direction was set to be perpendicular to the stripe patterns. The other four samples had isotropic surface geometry and the scanning angles did not matter. The scanning speed affected the tactile perception. The accuracy in perception was considered to be low when the scanning speed was either too high or too low. We decided the speed to be 4 cm/s, which is the typical speed of the finger when we attempt to capture the surface characteristics. The output voltage was recorded with a sampling time of 2 ms. This leads to a Nyquist frequency of 250 Hz. The slowly adaptive tactile receptors are most sensitive against the stimuli from 20 to 40 Hz, while the Pacinian corpuscle, which is one of the fast adaptive tactile receptors, is most sensitive at 200 Hz [8]. Although the fast adaptive receptors can detect the stimuli up to 1000 Hz, we consider that the signals below 250 Hz can be sufficiently effective. Encoding was repeated 100 times for each of the five samples while the scanned region of the sample surface was varied.

### 2.6. Tactile Presentation of Encoding Data

Ten participants (eight males and two females, aged 22 to 26 years) participated in the experiments. The encoded data were amplified 1000 fold after the median of the data was adjusted to 0 V. Since the larger amplification culminated in a better replication of the encoded surfaces in our prior work [11], we set the amplification rate to be 1000, which is limited by the experimental setup. First, the participants touched and recognized the surfaces of the five samples. Then, the participants scanned the display with the fingertip of their index finger at a speed of approximately 4 cm/s. The participants were asked whether the presented surface was similar to each of the five samples; they recorded 1 when it was similar and 0 when it was not. Note that the participants could select one or multiple samples or even none of them. For example, the evaluation vector was [1, 0, 0, 0, 1] when they considered the presented surface to be similar to the surfaces of wood and a paper towel. For a presentation surface that was similar to none of the samples, the vector was [0, 0, 0, 0, 0]. Fifty surfaces (10 for each sample) were presented to each participant. A total of 500 results were thus obtained from the ten participants. Although 500 is a relatively small number of results for conventional machine learning experiments, it is difficult to collect a large amount of data in tactile perception experiments (see Discussion section for an explanation).

This dataset represents the performance of the EP method. The sum of the vectors for each sample is given in Table 1. The values represent the replicability of the samples by the tactile display. For example, 79 out of the 100 presentations of a wood surface were perceived to be similar to a real wood surface. The performance of the EP method in terms of replicability was 84.8 ± 6.5%. The micropatterned polymer showed the highest replicability (97%), which may be due to its periodic and simple surface geometry.

As described above, the participants could select multiple samples that were similar to the presented surface. For example, as shown in Table 1, participants reported that 79 out of the 100 presentations of a wood surface were similar to real wood and that 29 and 32 of these presentations were also similar to the surfaces of PS foam and a paper towel, respectively. The specificity rate (i.e., the ratio of correct answers to the total number of answers) for the wood sample was 48.5% (79/163). The specificity rates for all samples are summarized in Table 2. The performance of the EP method in terms of specificity was 53.1 ± 5.3%.

## 3. Results

In this section, we describe the results of the machine learning and the signals produced to present the five material samples. We also discuss the reproduction of real tactile perception.

### 3.1. Deduction of Input Signals by Machine Learning

We first divided the 500 samples into 450 training samples and 50 validation samples. The MLP neural network consisted of a five-dimensional input layer, a 10-dimensional hidden layer, a 50-dimensional hidden layer, and a 250-dimensional output layer, as shown in Figure 4a. At the 75th epoch, the losses for the training and validation samples were reduced to 0.0075 and 0.0081, respectively (see Figure 4b). The results indicate that the model was well generalized. We admit that the smaller epoch numbers may be sufficient to generate the model though no significant overfitting was observed even at the 75th epoch. To deduce the signals for the presentation of the five samples, we input a five-dimensional one-hot vector to the trained model. For example, the signal for the wood sample was deduced by inputting [1, 0, 0, 0, 0] to the trained model. Figure 5 and Figure 6 show the encoded signals used for training (left column) and the signals deduced from the model (right column) in the time domain and the frequency domain for the five material samples. The horizontal and vertical axes show the frequency and absolute voltage before amplification, respectively. The deduced signals for the wood, micropatterned polymer, and paper towel samples, but not those for the lint-free wipe and PS foam samples, have frequency distributions similar to those for the encoded signals. The effects of these differences on perception are discussed in the following sections.

### 3.2. Tactile Presentation with Deduced Input Signals

The surfaces presented by the mechano-tactile display with the signals deduced from the model were evaluated by two groups of participants. The first group (Group A) had ten participants (eight males and two females, aged 20 to 26 years), each of which had participated in the perception experiments used to create the dataset (see Section 2.6). The second group (Group B) had ten participants (eight males and two females, aged 20 to 24 years), none of which had participated in the previous experiments. First, the participants touched and recognized the surfaces of the five samples. Then, the surfaces created based on the model for each sample were presented to the participants. The participants recorded the similarity of the presented surface to the sample for each surface. Ten five-dimensional evaluation vectors were obtained for each sample for each group. Table 3 and Table 4 show the sum of the evaluation vectors, which represents replicability, for Group A and Group B, respectively. Table 5 and Table 6 show the specificity rates for Group A and Group B, respectively. The results are summarized in Figure 7.

Under all conditions, the subjects tended to correctly identify the samples. The independence among the original samples was confirmed by a chi-squared test (*p* < 0.01). The replicability for all samples was 98 ± 4% for both Group A and Group B. This is significantly higher than that for the EP method (*p* < 0.05, Student *t*-test). The specificity rates were 64.2 ± 7.2% and 56.12 ± 9.4% for Group A and Group B, respectively. The specificity rate for Group A was significantly higher than that for the EP method (*p* < 0.05, Student *t*-test) while the specificity rate for Group B was similar to that for the EP method and the difference was not significant.

## 4. Discussion

This work attempted to solve the inverse problem for tactile displays, i.e., finding the input signals for a tactile display that presents physical sample surfaces. Even though the participants in the experiments were not required to select the most similar sample out of the five samples (i.e., they were allowed to select none of them), the replicability was 98 ± 4% for both Group A and Group B, which is significantly higher than the result for the EP method. We thus demonstrated that the proposed method can deduce the more appropriate input by fine-tuning the signals from the EP method.

The specificity rate for Group A, but not Group B, was significantly higher than that for the EP method. This is due to the small number of samples used for machine learning. It is difficult to collect a large amount of data in tactile perception experiments for the following reasons. First, in tactile experiments, the participants are asked to touch a tactile device that is unique in many cases. It is not possible to remotely obtain data from participants. Second, it takes participants some time to perceive tactile stimulation. These are the major differences compared to visual and audio perception experiments. Nevertheless, the replicability in the present study was high. The data were created based on the signals obtained via the EP method, which has been previously shown to have a replicability of over 80%. The machine learning fine-tuned the signals, which increased the replicability.

Regarding the lack of a large amount of data for tactile perception tests, we previously proposed applying natural language processing to the existing large language corpus instead of conducting perception experiments [17]. This approach has been used to categorize onomatopoeia related to tactile perception.

The training results show that the input signals for wood and paper towel were similar to each other. The relatively low accuracy for these samples may have originated from the properties of the tactile display and the characteristics of human tactile perception. As we discussed in our prior work [10], tactile displays are good at presenting some types of perception (e.g., roughness) but not good for others (e.g., wetness, warmth). The tactile properties that differentiate wood from a paper towel are considered to be the ones that tactile displays are not good at presenting. For lint-free wipe, PS foam, and micropatterned polymer samples, the accuracy was significantly higher (*p* < 0.05).

As shown in Figure 5 and Figure 6, the encoded data and the deduced signals are quite different for the lint-free wipe and PS foam and rather similar for wood, micropatterned polymer, and the paper towel. For the lint-free wipe, as shown in Figure 6b-1,b-2, the encoded signals have small peaks for almost all frequencies, whereas the signals deduced from machine learning show a characteristic peak at a low frequency. For the other cases, according to the signals in the frequency domain, machine learning enhanced the signals at characteristic frequencies and suppressed them at uncharacteristic frequencies. This is considered to enhance replicability.

## 5. Conclusions

In this study, machine learning was applied to solving the inverse problem for mechano-tactile displays. Applying machine learning for fine tuning improved the replicability of the EP method to over 90% (vs. 80% for the EP method alone). The improved replicability is attributed to machine learning enhancing characteristic frequency components and suppressing uncharacteristic frequency components. The proposed approach is applicable to the inverse problem for tactile displays. It can be used to develop applications that utilize tactile perception.

## Figures and Tables

**Figure 1 sensors-22-05299-f001:**
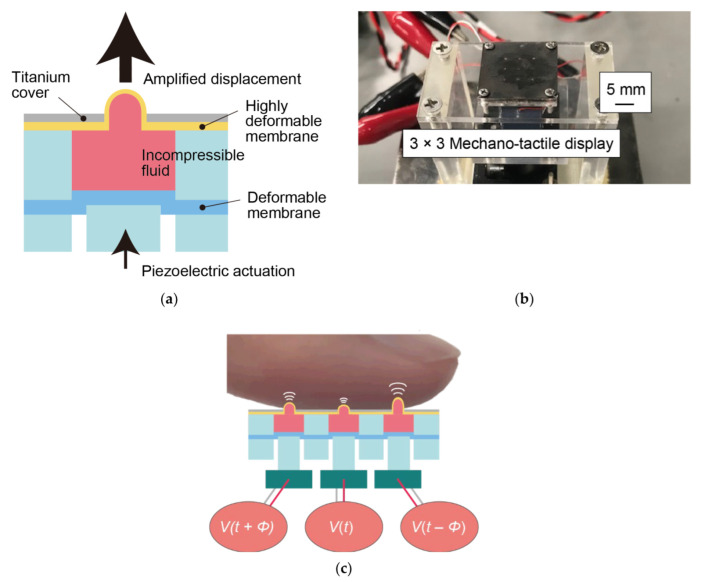
(**a**) Schematic diagram, (**b**) photograph, and (**c**) working principle of mechano-tactile display.

**Figure 2 sensors-22-05299-f002:**
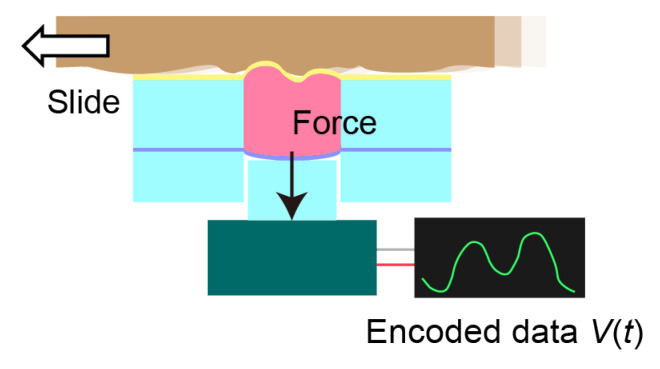
Encoding of samples.

**Figure 3 sensors-22-05299-f003:**
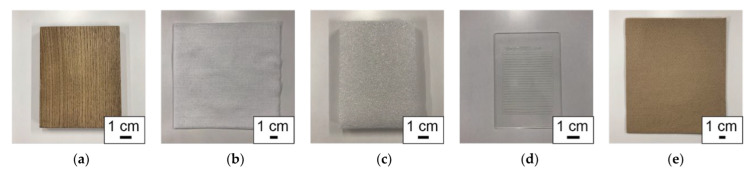
Samples used in this work. (**a**) Wood, (**b**) lint-free wipe, (**c**) PS foam, (**d**) micropatterned sample, and (**e**) paper towel.

**Figure 4 sensors-22-05299-f004:**
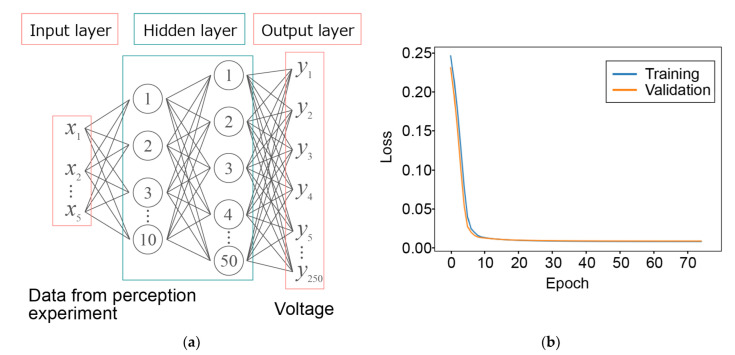
(**a**) Schematic diagram of MLP neural network. (**b**) Loss function versus epoch number for training and validation samples.

**Figure 5 sensors-22-05299-f005:**
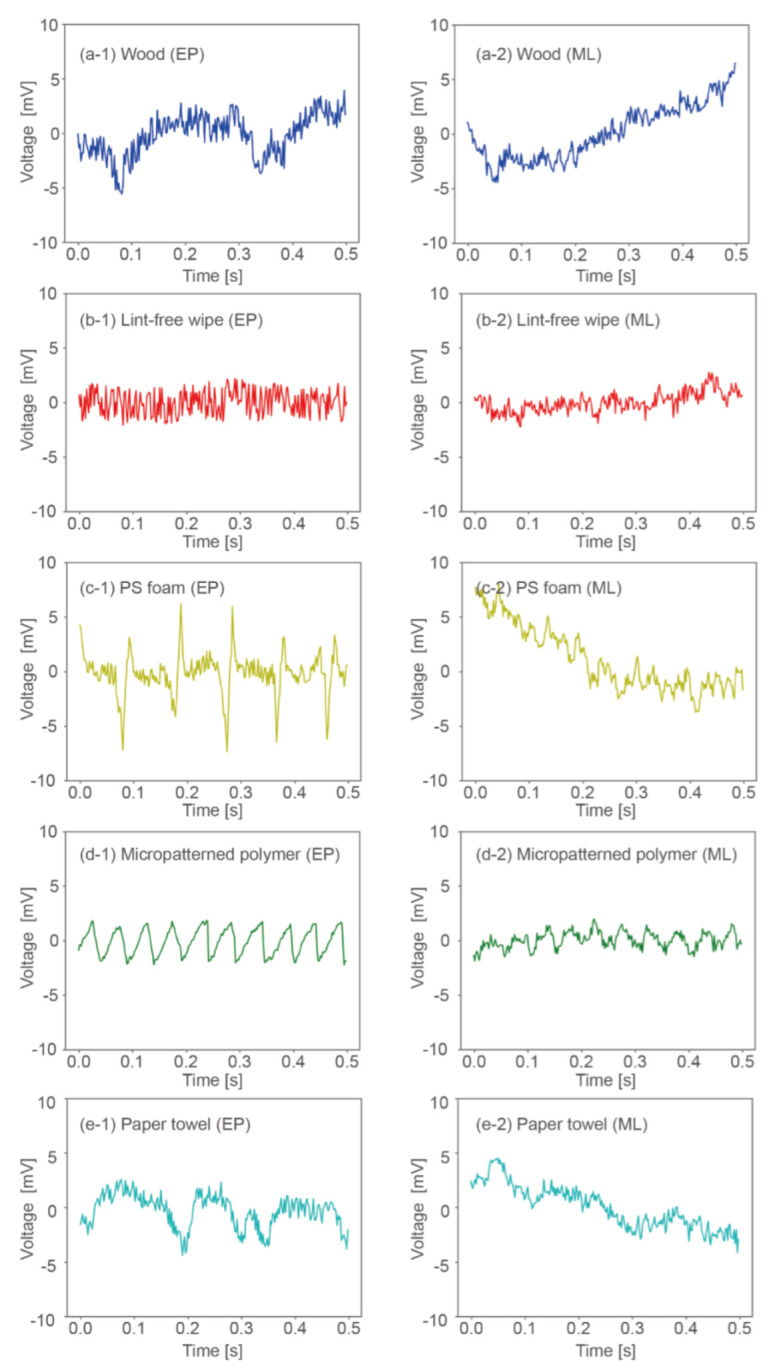
Waveforms deduced by EP method and machine learning (ML) for (**a**) wood, (**b**) lint-free wipe, (**c**) PS foam, (**d**) micropatterned polymer, and (**e**) paper towel.

**Figure 6 sensors-22-05299-f006:**
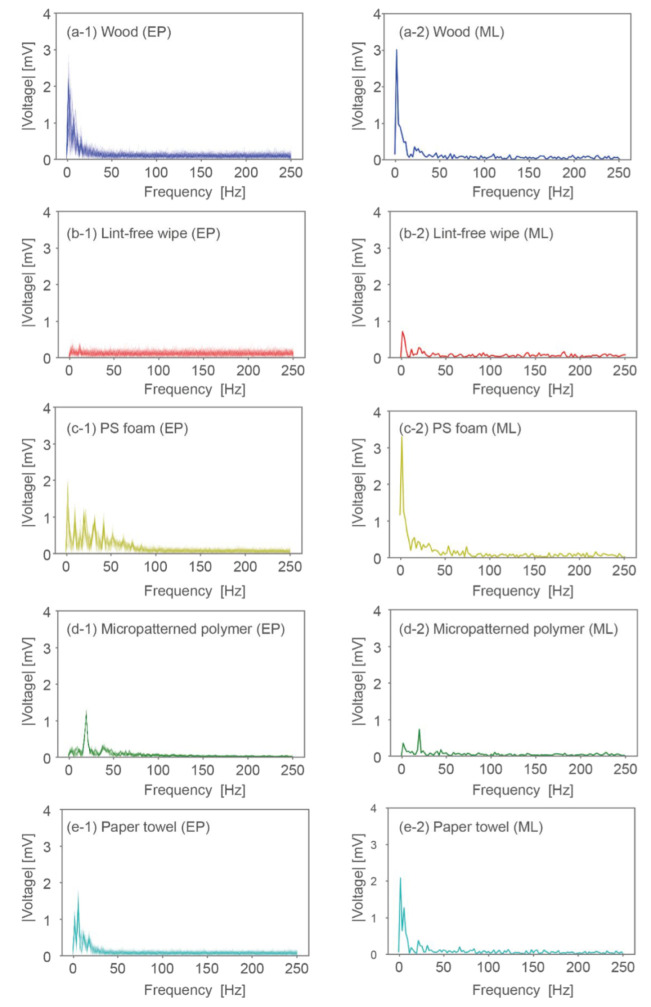
Frequency component of waveforms deduced by EP method and machine learning (ML) for five samples.

**Figure 7 sensors-22-05299-f007:**
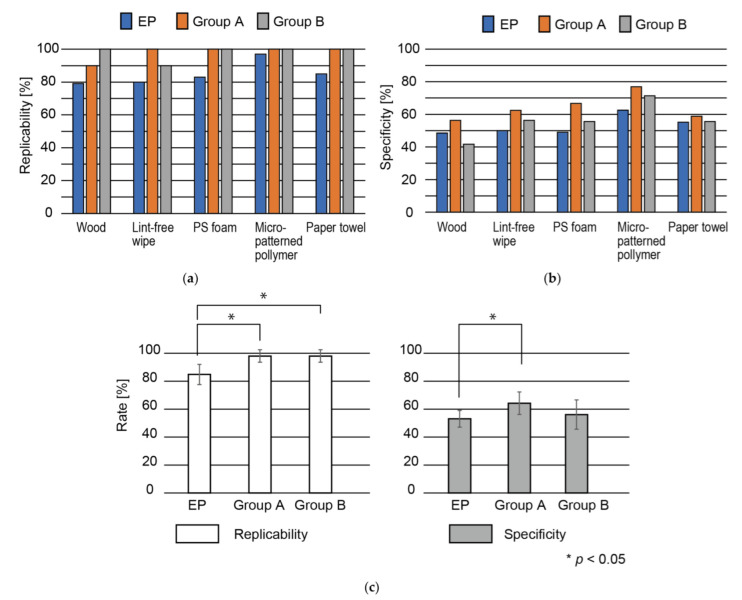
(**a**) Replicability rates and (**b**) specificity rates for approach that combines EP method and machine learning for Group A and Group B for all samples. (**c**) Average replicability rates and specificity rates among all samples for EP method with machine learning for Group A and Group B.

**Table 1 sensors-22-05299-t001:** Sum of 100 five-dimensional vectors as a percentage for the surfaces display based on EP method for wood, lint-free wipe (LF wipe), PS foam, micropatterned polymer (M-poly), and paper towel (Towel).

Sample	Wood	LF Wipe	PS Foam	M-poly	Towel
**Wood**	79	19	29	4	32
**LF wipe**	26	80	19	8	27
**PS foam**	35	22	83	7	22
**M-poly**	16	12	13	97	17
**Towel**	24	18	22	5	85

**Table 2 sensors-22-05299-t002:** Specificity rates as a percentage for the surfaces display based on EP method for wood, lint-free wipe (LF wipe), PS foam, micropatterned polymer (M-poly), and paper towel (Towel).

Sample	Wood	LF Wipe	PS Foam	M-poly	Towel
**Wood**	48.5	11.7	17.8	2.5	19.6
**LF wipe**	16.3	50.0	11.9	5.0	16.9
**PS foam**	20.7	13.0	49.1	4.1	13.0
**M-poly**	10.3	7.7	8.4	62.6	11.0
**Towel**	15.6	11.7	14.3	3.2	55.2

**Table 3 sensors-22-05299-t003:** Replicability for the model-based surface presentation for Group A.

Sample	Wood	LF Wipe	PS Foam	M-poly	Towel
**Wood**	9	1	1	1	4
**LF wipe**	1	10	3	0	2
**PS foam**	3	1	10	0	1
**M-poly**	1	1	0	10	1
**Towel**	3	2	1	1	10

**Table 4 sensors-22-05299-t004:** Replicability for the model-based surface presentation for Group B.

Sample	Wood	LF Wipe	PS Foam	M-poly	Towel
**Wood**	10	4	3	1	6
**LF wipe**	3	9	3	0	1
**PS foam**	3	3	10	0	2
**M-poly**	1	1	1	10	1
**Towel**	4	1	2	1	10

**Table 5 sensors-22-05299-t005:** Specificity rates as a percentage for the model-based surface presentation for Group A.

Sample	Wood	LF Wipe	PS Foam	M-poly	Towel
**Wood**	56.3	6.3	6.3	6.3	25.0
**LF wipe**	6.3	62.5	18.8	0.0	12.5
**PS foam**	20.0	6.7	66.7	0.0	6.7
**M-poly**	7.7	7.7	0.0	76.9	7.7
**Towel**	17.6	11.8	5.9	5.9	58.8

**Table 6 sensors-22-05299-t006:** Specificity rates as a percentage for the model-based surface presentation for Group B.

Sample	Wood	LF Wipe	PS Foam	M-poly	Towel
**Wood**	41.7	16.7	12.5	4.2	25.0
**LF wipe**	18.8	56.3	18.8	0.0	6.3
**PS foam**	16.7	16.7	55.6	0.0	11.1
**M-poly**	7.1	7.1	7.1	71.4	7.1
**Towel**	22.2	5.6	11.1	5.6	55.6

## Data Availability

The data presented in this study are openly available in FigShare at at doi:10.6084/m9.figshare.20097392.

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
