# Peer review of "Machine-Learning-Based Fine Tuning of Input Signals for Mechano-Tactile Display"

_sensors, 2022, doi:10.3390/s22145299_

Round 1
Reviewer 1 Report
This paper presented a novel deducing input signal method that can present the target surface by training a multilayer perceptron with the subjective evaluation of tactile sensation and the input signals. The similarity of the surface compared to five material samples is recorded with a psychophysical experiment. The deduced new inputs are evaluated with the target surface. The significantly better accuracy in deducing the inputs of five material samples proves the proposed method. The paperwork is of great importance for the inverse problem of tactile displays.
Some comments:
[1] A dataset of 500 vectors was collected for the proposed machine learning method, where each of the five samples was perceived 100 times for the various region. What are the differences between the 100 samples for each sample material? Should it be better to scan each sample with different speeds and different angles over the same region of the sample, because the surface texture of each sample is mightly not isotropic?
[2] During the process of obtaining the tactile perception of samples, should the moving speed (a speed of approximately 4 cm/s was mentioned in the manuscript) of the participant’s finger have a prominent influence on the performance of the experiment?
[3] In “3.2. Tactile Presentation with Deduced Input Signals”, two groups of ten participants were involved in the experiments. Have the two groups been asked to perceive the real five samples before the experiment? Or they just give their answer based on their experience of touching various objects in their daily lives.
[4] The experiments show that machine learning for fine-tuning leads to significantly better accuracy for tactile display. What are the main reasons for these improvements? Are the machine learning-based inputs more similar to the real surface properties of the samples, or just making the simulated surface more perceivable to human subjects, like zooming-in effects in visual rendering.
[5] Should the last author in Ref 9 and Ref11 be “Miki, N” instead of “Miki, M”?
Author Response
This paper presented a novel deducing input signal method that can present the target surface by training a multilayer perceptron with the subjective evaluation of tactile sensation and the input signals. The similarity of the surface compared to five material samples is recorded with a psychophysical experiment. The deduced new inputs are evaluated with the target surface. The significantly better accuracy in deducing the inputs of five material samples proves the proposed method. The paperwork is of great importance for the inverse problem of tactile displays.
Some comments:
[1] A dataset of 500 vectors was collected for the proposed machine learning method, where each of the five samples was perceived 100 times for the various region. What are the differences between the 100 samples for each sample material? Should it be better to scan each sample with different speeds and different angles over the same region of the sample, because the surface texture of each sample is mightly not isotropic?
Thank you for your comment. As we describe in 2.5., Encoding was repeated 100 times for each of the five samples while the scanned region of the sample surface was varied. The micropatterned polymer has a periodical stripe pattern and the scanning angle is critical in perception. We set the scanning direction to be perpendicular to the stripe patterns. The other 4 samples have rather isotropic and uniform surface geometries. Changing the scanning location and scanning angle would lead to similar results. The scanning speed is important in tactile perception. The accuracy of the tactile perception is low when the scanning speed is either extremely fast or slow. We asked the participants to scan the surface at a speed of approximately 4 cm/s, which is a typical scanning speed when we move the finger to identify the surface. We added these points in 2.5. Encoding of Material Surfaces.
2.5. Encoding of Material Surfaces, Line 203:
In the case of the micropatterned sample, the scanning direction was set to be perpendicular to the stripe patterns. The other 4 samples have isotropic surface geometry and the scanning angles do not matter. The scanning speed affects the tactile perception. The accuracy in perception is considered to be low when the scanning speed is either too high or too low. We decided the speed to be 4 cm/s, which is the typical speed of the finger when we attempt to capture the surface characteristics.
[2] During the process of obtaining the tactile perception of samples, should the moving speed (a speed of approximately 4 cm/s was mentioned in the manuscript) of the participant’s finger have a prominent influence on the performance of the experiment?
We consider that the answer is “yes”, though we have not investigated the effect of the scanning speed thoroughly. We decided to make the scanning speed to be similar among the experiments. We mentioned this in 2.5. Encoding of Material Samples (also see the response for [1]).
[3] In “3.2. Tactile Presentation with Deduced Input Signals”, two groups of ten participants were involved in the experiments. Have the two groups been asked to perceive the real five samples before the experiment? Or they just give their answer based on their experience of touching various objects in their daily lives.
Thank you for your comment. At the beginning of the experiments, the participants touched the material samples and recognized them. We added the explanation in 3.2.
3.2. Tactile Presentation with Deduced Input Signals, Line 283.
The surfaces presented by the mechano-tactile display with the signals deduced from the model were evaluated by two groups of participants. The first group (Group A) had ten participants (eight males and two females, aged 20 to 26 years), each of which had participated in the perception experiments used to create the dataset (see Sec. IV-C). The second group (Group B) had ten participants (eight males and two females, aged 20 to 24 years), none of which had participated in the previous experiments. First, the participants touched and recognized the surfaces of the five samples. Then, the surfaces created based on the model for each sample were presented to the participants.
[4] The experiments show that machine learning for fine-tuning leads to significantly better accuracy for tactile display. What are the main reasons for these improvements? Are the machine learning-based inputs more similar to the real surface properties of the samples, or just making the simulated surface more perceivable to human subjects, like zooming-in effects in visual rendering.
Thank you for your precious comments. As we described in 4. Discussion, the signals are different.
As shown in Figures 5 and 6, the encoded data and the deduced signals are quite different for the lint-free wipe and PS foam and rather similar for wood, micropatterned polymer, and the paper towel. For the lint-free wipe, as shown in Figures 6 (b-1) and (b-2), the encoded signals have small peaks for almost all frequencies, whereas the signals deduced from machine learning show a characteristic peak at low frequency. For the other cases, according to the signals in the frequency domain, machine learning enhanced the signals at characteristic frequencies and suppressed them at uncharacteristic frequencies. This is considered to enhance replicability.
However, we cannot conclude how the differences in the signals lead to the accuracy of the tactile perception yet. Our hypothesis is the same as your second opinion, “making the simulated surface more perceivable to human subjects, like zooming-in effects in visual rendering”. We will look into this important issue in future work.
[5] Should the last author in Ref 9 and Ref11 be “Miki, N” instead of “Miki, M”?
They are Miki, N. Thank you.
Reviewer 2 Report
The manuscript entitled “Machine-Learning-Based Fine Tuning of Input Signals for Mechano-Tactile Display”
The authors have encoded the target surface by scanning it using an array of piezoelectric device. The authors have reported the machine learning which is promising method for solving inverse problems and the replicability is better than their previous work (encoding and presentation, EP). However, few points are still not clear in this review. Therefore, I recommend the paper for the publication after minor modification.
1. The authors have used five material samples for the experiment. It would be great to give explain why they are interested in those five materials.
2. Please explain why the replicability of their proposed method is significantly higher than the results for the EP method.
Author Response
The authors have encoded the target surface by scanning it using an array of piezoelectric device. The authors have reported the machine learning which is promising method for solving inverse problems and the replicability is better than their previous work (encoding and presentation, EP). However, few points are still not clear in this review. Therefore, I recommend the paper for the publication after minor modification.
- The authors have used five material samples for the experiment. It would be great to give explain why they are interested in those five materials.
Thank you for your comment. First of all, we would like to note that there are no standard samples for tactile experiments and researchers tend to select the samples that are common and easy to obtain. In our prior work (Y. Kosemura et al., Japanese Journal of Applied Physics, 53, 06JM11 (2014), M. Kawazoe et al., Sensors and Actuators A: Physical, 261, 30–39 (2017)), we used 20 and 18 samples as the tactile samples, which include wood, polystyrene form, wipe, and towel. In addition to these four samples, we used micropatterned polymer as the sample, which was proposed as the microfabricated tactile sample that can vary the surface geometry, surface chemistry, and material property (stiffness, thermal conductivity, etc.) using microfabrication technologies. We added the following explanation in 2.4. Data Collection.
2.4. Data Collection, Line 169:
In this work, we used five material samples, namely wood, a lint-free wipe, PS foam, a micropatterned polymer, and a paper towel (see Figure 3). The wood, the lint-free wipe, the PS foam, and the paper towel samples are commercially available and were used as the tactile samples in our prior work [11]. The micropatterned polymer was proposed as the microfabricated tactile sample, which can vary the surface geometry, surface chemistry, and the material properties (stiffness, thermal conductivity, etc.) using microfabrication technologies [11]. First, these five samples were encoded and presented to participants using the mechano-tactile display.
- Please explain why the replicability of their proposed method is significantly higher than the results for the EP method.
In 4. Discussion, we describe the reason as,
As shown in Figures 5 and 6, the encoded data and the deduced signals are quite different for the lint-free wipe and PS foam and rather similar for wood, micropatterned polymer, and the paper towel. For the lint-free wipe, as shown in Figures 6 (b-1) and (b-2), the encoded signals have small peaks for almost all frequencies, whereas the signals deduced from machine learning show a characteristic peak at low frequency. For the other cases, according to the signals in the frequency domain, machine learning enhanced the signals at characteristic frequencies and suppressed them at uncharacteristic frequencies. This is considered to enhance replicability.
In our future work, we highlight and investigate the signals at the characteristic frequencies. Our hypothesis is that these enhanced signals can be related to the characteristics of the tactile receptors and/or the cognitive processes of the receptors' output. However, it is not conclusive yet and we do not describe it in this manuscript.
Reviewer 3 Report
The authors present a novel tactile display that is capable of simulating different materials. To this end, they use machine learning to train the display.
The contribution of the paper to the field is very significant. The main concern is that the loss function curve seems to show some degree of overfitting: some more detailed explanation would be required.
Minor English corrections are required (e.g., "3. Experimental" on page 6 should probably be "3. Experimental study")
Author Response
The authors present a novel tactile display that is capable of simulating different materials. To this end, they use machine learning to train the display.
The contribution of the paper to the field is very significant. The main concern is that the loss function curve seems to show some degree of overfitting: some more detailed explanation would be required.
Thank you for your comment. We carefully looked into the loss function in Figure 4(b) and we consider that no significant overfitting took place. However, we must admit that the number of epochs can be fewer (perhaps, 25 is good enough). We added the following explanation to the manuscript.
3.1. Deduction of Input Signals by Machine Learning, Line 268:
We admit that the smaller epoch numbers may be sufficient to generate the model though no significant overfitting was observed even at the 75th epoch.
Minor English corrections are required (e.g., "3. Experimental" on page 6 should probably be "3. Experimental study")
We have asked for an English editing service.